# Plant Diversity and Aboveground Biomass Interact with Abiotic Factors to Drive Soil Organic Carbon in Beijing Mountainous Areas

**Piao Zhou, Lin Zhang and Shi Qi ***





School of Soil and Water Conservation, Beijing Forestry University, Beijing 100083, China
* Correspondence: qishi@bjfu.edu.cn

**Abstract:** We analyzed and compared the effects of biotic factors (species diversity, structural diversity, and aboveground biomass) and abiotic factors (topography and soil properties) on soil organic carbon in selected mountainous areas of Beijing China. An overall goal is to provide a preliminary scientific basis for biodiversity protection and coordinated development of forest ecosystems and the subsequent carbon balance in this region. Our study ecosystems were coniferous forests, mixed coniferous and broadleaved forests, and broadleaved forests in the western mountainous area of Beijing. We determined both direct and indirect effects of abiotic and biotic factors on soil organic carbon using multiple linear regression and a structural equation model. Results showed that the biotic factors aboveground biomass and structural diversity were the main driving forces of organic carbon accumulation in the soil surface layer (0–10 cm), but species diversity had no significant effect ($p > 0.05$). Abiotic factors (altitude, total nitrogen, and total potassium) had some influence on soil surface organic carbon but were subordinate to the biotic factors. The biotic factors had no significant effect on soil organic carbon in the subsurface layers (10–20 cm) and (20–30 cm) ($p > 0.05$), whereas the abiotic factors altitude and total nitrogen played a dominant role in subsurface soil organic carbon accumulation of both layers ($p < 0.01$). The influences were both direct and indirect influences, but the direct influences played a major role. Our results form an initial reference for afforestation management (Beijing-Tianjin Sandstorm-source Control Project) from the perspective of biodiversity protection and carbon storage enhancement.

**Keywords:** biotic factors; abiotic factors; soil organic carbon; multiple linear regression; structural equation model

## 1. Introduction

Soil represents the largest carbon pool in the forest ecosystem. Due to the influence of climate change and human activities, the soil carbon of the forest ecosystem has been lost a lot in recent decades; therefore, soil carbon sequestration plays an important role in the global carbon cycle and climate change regulation [1–4]. A core issue in understanding the formation mechanism of soil carbon sink and a controversial topic is how plant diversity and aboveground biomass drive the formation of soil organic carbon [5,6]. Generally speaking, the accumulation mechanism of soil organic carbon subtly regulates the formation of soil organic carbon mainly through factors such as plant root exudates [7], root contribution [8], litter decomposition [9], and competition between plants and decomposers [10]. Recent small-scale studies have shown that plant diversity can increase soil organic carbon storage by increasing underground carbon inputs [11], microbial activities, and inhibiting carbon loss during decomposition [12]. However, many large-scale studies have demonstrated that the formation pattern of soil organic carbon storage was controlled by a wider range of climate [13], topography, vegetation types, soil, and other factors [14,15]. Therefore, how biotic factors (plant diversity, aboveground biomass) and abiotic factors

commonly drive the formation of soil organic carbon and the relative importance of these driving factors deserve further study in the forest ecosystem.

Many scholars have focused on the driving factors of soil organic carbon, but there is no consensus on its formation mechanism. Cao et al. [16] pointed out that higher species diversity promotes the formation of soil organic carbon by increasing underground carbon inputs and microbial community diversity; Wang et al. [17] also found that species diversity may promote the formation of soil organic carbon through an interspecific competition based on outdoor planting experiments. The ecological niche complementation effect is the main mechanism for explaining the above research. This effect suggests that higher species diversity can promote the growth of forest trees [18], ecosystem stability, and soil organic carbon by improving the utilization efficiency of light, heat, water, and fertilizer resources by plant storage [19]. However, in complex forest ecosystems, higher species diversity may inhibit the formation of organic carbon because plants over-repress microbial activity by absorbing nitrogen and mineral elements from the soil. In addition, due to the influence of abiotic factors such as climate, topography, and soil, the storage of soil organic carbon is also reduced to a certain extent [20,21]. Structural diversity is used as an indicator to characterize the spatial heterogeneity of forest trees. Related studies have shown that the driving mechanism of soil organic carbon may be related to structural diversity, and the complex stand structure may accelerate the formation of organic carbon. Zhang et al. [22] believed that the litter C/N ratio under the complex structure affected the accumulation of soil organic carbon by changing the number of soil microorganisms and the activity of soil invertases. However, the relationship between structural diversity and soil organic carbon depends on the influence of abiotic factors such as climate, topography, and soil under special circumstances [23]. Laurent et al. [24] have suggested that the relationship between soil organic carbon and plant diversity also depends on aboveground biomass, and species diversity can promote the formation of organic carbon only when there is an excess of aboveground biomass. To sum up, it is difficult to fully reveal the true mechanism of soil organic carbon formation by only attributing the driving mechanism of soil organic carbon formation to species diversity or structural diversity while ignoring the interaction between aboveground biomass and abiotic factors. Therefore, when studying the mechanism of plant diversity on soil organic carbon formation, aboveground biomass and abiotic factors should be comprehensively considered, and the relative contribution of each driving factor to soil organic carbon should be comprehensively revealed.

Beijing-Tianjin sandstorm source afforestation project plays a significant role in alleviating local climate warming and improving the ecological environment in Beijing mountainous areas; the main purpose of the project is to improve the stand structure, plant diversity, and carbon storage of the forest ecosystem. Present experimental studies on the driving mechanism of soil organic carbon in afforestation projects mainly focus on different vegetation types, stand structures, and site conditions, while the research on the relative influence mechanism of plant diversity, aboveground biomass, and abiotic factors on the formation of soil organic carbon is relatively lacking. On this basis, the study took coniferous forests, mixed coniferous and broadleaved forests, and broadleaved forests in the western mountainous area of Beijing as the objects, and based on the measured data of 48 standard plots, the coupling driving mechanism of biotic factors (species diversity, structural diversity, aboveground biomass) and abiotic factors to soil organic carbon was expounded by using multiple linear regression and structural equation model, so as to provide theoretical and technical support for the improvement of forest quality in mountainous areas of Beijing.

## 2. Materials and Methods

### 2.1. Study Sites

The research area (39°12′–41°05′ N, 115°25′–116°06′ E) was located in the Beijing-Tianjin sandstorm source project area for closing hillsides for afforestation, mainly in the western mountainous area of Beijing, including Fangshan, Mentougou, Changping,

and Yanqing (Figure 1). The climate was mainly temperate monsoon climate, with an average annual rainfall of 480–620 mm and an average annual temperature of 11–13 °C. The soil types were mainly brown forest soil and mountain cinnamon soil. Vegetation types include coniferous tree species such as *Platycladus orientalis* and *Pinus tabulaeformis*, broadleaved tree species such as *Cotinus coggygria*, *Prunus armeniaca*, *Ulmus pumila*, and *Oak*, while undergrowth herbaceous plants are mainly *Artemisia stechmanniana*, *Cryptosporidium fasciculatum*, *Carex rigescens*, *Potentilla chinensis*, and *Selaginella sinensis*.

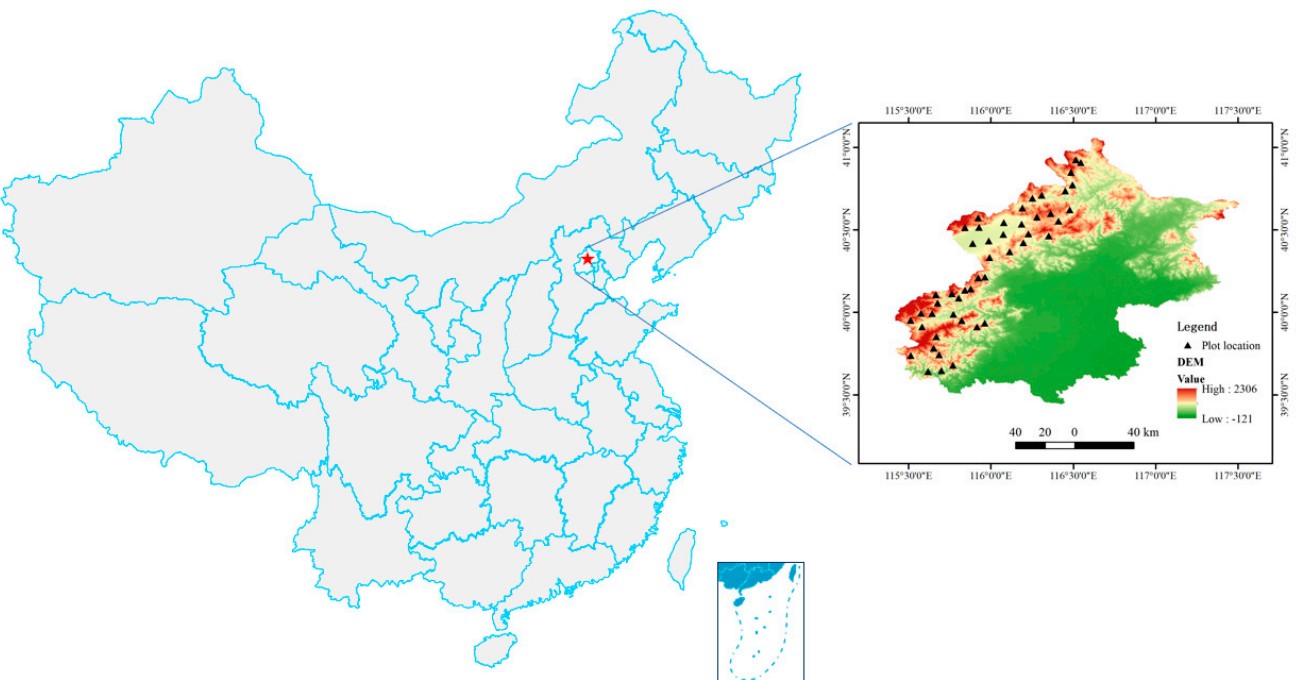

**Figure 1.** Geographical map of the study area.

### 2.2. Data Sampling

In June 2021, coniferous forests, mixed coniferous and broadleaved forests, and broadleaved forests with less human disturbance and better growth conditions were selected as the research objects in the study area, and 48 standard plots (20 m × 20 m) were established. We investigated the basic stand structure factors such as height, DBH, and stand density of all tree species in the standard sample plot, three 5 m × 5 m shrub and 1 m×1 m herbaceous quadrats were set in the standard quadrats, and the biomass of shrub and grass in the quadrats was investigated and sampled, the litter was collected as the source of aboveground biomass data in the quadrats. At the same time, the abiotic factors such as elevation and slope of standard plots were measured by hand-held GPS and slope meter. In addition, three representative soil sampling points were randomly selected in each standard sample plot, and the soil samples were collected in three layers (0–10 cm, 10–20 cm, and 20–30 cm) with a standard ring knife of 100 cm$^3$. Each layer of soil samples was repeated three times, producing a total of 432 samples. Sample preparation involved air drying, mixing, crushing, and sieving (2 mm). Laboratory analysis included organic carbon, total nitrogen, total potassium, total phosphorus, and pH. Total nitrogen soil was determined by the Kjeldahl analysis via an infrared N furnace. Total phosphorus was determined from a $HClO_4$-$H_2SO_4$ dissolution by spectrophotometer. Total potassium was determined from a $HNO_3$-$HClO_4$ digestion by atomic absorption. Soil pH was determined in 1:1 soil water suspension. Organic carbon was determined by the Walkley Black method [25].

*2.3. Data Analyses and Modeling*

2.3.1. Plant Diversity Index

In this study, the Shannon diversity index, Simpson diversity index, and Margalef richness index were selected to represent the species diversity level of sample plots. Because the DBH and height of tree species in sample plots were easy to measure accurately, the structural diversity level of sample plots could be reflected by calculating the tree height, DBH Shannon index, and Simpson index. The calculation results of indexes are shown in Table 1.

**Table 1.** Calculation formula and results of each index.

| Index | Formula | Mean | Range |
|---|---|---|---|
| Species Shannon index | $H_s = -\sum\limits_{i=1}^{N_s} \frac{n_i}{N} \times ln\left(\frac{n_i}{N}\right)$ | 1.13 | 0.12–2.01 |
| Species Simpson index | $D_s = 1 - \sum\limits_{i=1}^{N_s} \left(\frac{n_i}{N}\right)^2$ | 0.59 | 0.05–0.84 |
| Species Margalef index | $S = (N_s - 1)lnN$ | 1.09 | 0.27–2.66 |
| DBH Shannon index | $H_d = -\sum\limits_{j=1}^{N_d} \frac{N_d}{N} \times ln\frac{n_j}{N}$ | 1.79 | 1.12–2.21 |
| DBH Simpson index | $D_d = 1 - \sum\limits_{j=1}^{N_d} \left(\frac{n_j}{N}\right)^2$ | 0.81 | 0.62–0.94 |
| Tree height Shannon index | $H_h = -\sum\limits_{k=1}^{N_h} \frac{n_k}{N} \times ln\left(\frac{n_k}{N}\right)$ | 1.57 | 0.94–2.06 |
| Tree height Simpson index | $D_h = 1 - \sum\limits_{k=1}^{N_h} \left(\frac{n_k}{N}\right)^2$ | 0.76 | 0.56–0.88 |

Notes: $N$ is the total number of trees, $N_s$ is the species of trees, $n_i$ is the number of the ith tree, and $N_d$ is the DBH series; $n_j$ is the number of the *j*th DBH level; $N_h$ is the number of tree levels; $n_k$ is the number of the *k*th tree level.

2.3.2. Aboveground Biomass

The biomass of the tree layer was calculated according to the regression equation of biomass-volume estimation established by Fang et al. [26]. The biomass of the understory shrub-grass layer was harvested, and the fresh weight samples of shrubs, herbs, and litter were taken back to the laboratory and dried at 70 °C to constant weight so as to obtain the biomass of the shrub-grass layer in each square.

2.3.3. Multiple Linear Regression Analysis

As soil organic carbon was comprehensively influenced by many factors, it could be analyzed by multiple linear regression with soil organic carbon as a dependent variable and species diversity and structural diversity indicators as independent variables:

$$SOC = a_0 + a_1 D_i + a_2 D_j + \varepsilon \tag{1}$$

where *SOC* means soil organic carbon, $a_0$ means intercept, $a_1$ and $a_2$ mean the explanatory coefficient of structural diversity and species diversity indicators, $D_i$ and $D_j$ mean structural diversity and species diversity indicators, and $\varepsilon$ means error term.

2.3.4. Structural Equation Model

A structural equation model was used to simulate and verify the causal relationship between biotic factors and abiotic factors, and soil organic carbon content in each soil layer. Before constructing the structural equation model, the Pearson correlation coefficient method in SPSS was used to test the correlation between aboveground biomass and abiotic factors and soil organic carbon in each soil layer, and the influential factors with high significance were selected. Combined with the species diversity and structural diversity index in the optimal multiple regression model, these factors were used as the basic variables in the construction of the SEM model.

The accuracy of the model was reflected by the fitting index of the model. In this study, the CMIN/DF, CFI, and RSMEA were selected to test the goodness of the model. Generally speaking, when the CMIN/DF was between 1 and 3, the RMSEA value was less than 0.08, and the CFI was greater than 0.9, which indicates that the fitting degree of the assumed model was high.

## 3. Results

The multiple linear regression analysis results for the relationship between plant diversity indexes and soil organic carbon are shown in Table 2. For a soil layer of 0–10 cm, the model with Simpson index of species and Shannon index of tree height as explanatory variables had the highest goodness; for the soil layer of 10–20 cm, the model with Shannon index and Simpson index of DBH as explanatory variables had the highest goodness; for the soil layer of 20–30 cm, the model with Simpson index and Shannon index of DBH as explanatory variables had the highest goodness. In the regression model, the index of community structure diversity was negatively correlated with the content of soil organic carbon in each soil layer. However, for the soil layers of 10–20 cm and 20–30 cm, the Shannon index and Simpson index of DBH had low explanations for the variation of soil organic carbon content, indicating that the diversity of community structure was not the dominant factor driving the formation of soil organic carbon.

The Pearson correlation analysis results for the relationship between aboveground biomass, altitude, slope, total nitrogen, total potassium, and soil organic carbon content in each soil layer are shown in Table 3. Aboveground biomass, altitude, total nitrogen, and total potassium were significantly related to soil organic carbon content of 0–10 cm, while the content of soil organic carbon in 10–20 cm and 20–30 cm soil layers was only related to abiotic factors such as altitude, total nitrogen, and pH. Therefore, when analyzing the driving mechanism of 0–10 cm soil organic carbon, the aboveground biomass, altitude, total nitrogen, and total potassium could be added to the structural equation model as variables. For the soil layer of 10–20 cm, the altitude and total nitrogen were added into the structural equation model as variables; for the soil layer of 20–30 cm, the formation of soil organic carbon was mainly related to abiotic factors, the effect of biotic factors such as species diversity, structural diversity, and aboveground biomass on soil organic carbon was no longer considered.

The optimal structural equation model (CMIN/DF = 1.194, CFI = 0.995, RSMEA = 0.048) of the driving factors of soil organic carbon (0–10 cm) was shown in Figure 2. which indicates that the model had good goodness of fit, and could be used to test the coupling mechanism of biotic and abiotic factors on soil organic carbon, Simpson index had no significant influence on Shannon index of tree height, aboveground biomass and soil organic carbon ($p > 0.05$), with path coefficients of −0.19, 0.09 and −0.04, respectively; Shannon index of tree height had a significant effect on aboveground biomass and soil organic carbon ($p < 0.05$), with path coefficients of −0.56 and 0.18, respectively; the aboveground biomass had a significant effect on soil organic carbon ($p < 0.001$), with path coefficient of 0.19. At the same time, abiotic factors had a significant effect on the Shannon index of tree height and soil organic carbon storage ($p < 0.05$), with path coefficients of −0.67 and 0.98, respectively, but no significant effect on aboveground biomass ($p > 0.05$), with path coefficient of 0.17. For the soil layer of 10–20 cm, the optimal structural equation model of the driving factors of soil organic carbon was shown in Figure 3. When only abiotic factors were combined to analyze the effect of species diversity and structural diversity on soil organic carbon, the Shannon index and Simpson index of DBH had no significant effect on soil organic carbon ($p > 0.05$), with path coefficients of −0.03 and 0.04, respectively. Meanwhile, the Shannon index of species did not significantly affect the Simpson index of DBH ($p > 0.05$), with a path coefficient of −0.11, and abiotic factors had a significant impact on soil organic carbon ($p < 0.01$), with a path coefficient of 0.98, which had a significant impact on Simpson index of DBH ($p < 0.05$), and a path coefficient was −0.37.

**Table 2.** Multiple regression analysis between species diversity, structural diversity, and soil organic carbon content in each soil layer.

| Depth of Soil Laye(cm) | Explanatory Variables | a1 | a2 | R2 |
|---|---|---|---|---|
| 0–10 | Hs + Hd | 0.049 | −0.586 | 0.353 |
| | Hs + Dd | 0.060 | −0.557 | 0.320 |
| | Hs + Hh | −0.092 | −0.628 | 0.364 |
| | Hs + Dh | −0.040 | −0.545 | 0.286 |
| | Ds + Hd | 0.090 | −0.592 | 0.351 |
| | Ds + Dd | 0.046 | −0.564 | 0.346 |
| | Ds + Hh | −0.111 | −0.623 | 0.368 |
| | Ds + Dh | −0.080 | −0.550 | 0.291 |
| | S + Hd | 0.045 | −0.586 | 0.352 |
| | S + Dd | 0.040 | −0.557 | 0.318 |
| | S + Hh | −0.087 | −0.627 | 0.364 |
| | S + Dh | −0.020 | −0.539 | 0.285 |
| 10–20 | Hs + Hd | −0.089 | −0.271 | 0.076 |
| | Hs + Dd | −0.091 | −0.329 | 0.110 |
| | Hs + Hh | −0.093 | −0.107 | 0.014 |
| | Hs + Dh | −0.107 | −0.172 | 0.030 |
| | Ds + Hd | −0.041 | −0.262 | 0.070 |
| | Ds + Dd | −0.028 | −0.319 | 0.103 |
| | Ds + Hh | −0.055 | −0.089 | 0.005 |
| | Ds + Dh | −0.067 | −0.155 | 0.024 |
| | S + Hd | −0.066 | −0.270 | 0.072 |
| | S + Dd | −0.079 | − 0.332 | 0.108 |
| | S + Hh | −0.061 | −0.097 | 0.009 |
| | S + Dh | −0.071 | −0.160 | 0.025 |
| 20–30 | Hs + Hd | 0.021 | −0.249 | 0.063 |
| | Hs + Dd | 0.026 | −0.232 | 0.056 |
| | Hs + Hh | 0.015 | −0.102 | 0.012 |
| | Hs + Dh | 0.008 | −0.143 | 0.021 |
| | Ds + Hd | 0.083 | −0.249 | 0.070 |
| | Ds + Dd | 0.095 | −0.237 | 0.064 |
| | Ds + Hh | 0.068 | −0.091 | 0.016 |
| | Ds + Dh | 0.062 | −0.133 | 0.025 |
| | S + Hd | 0.074 | −0.261 | 0.068 |
| | S + Dd | −0.075 | −0.247 | 0.061 |
| | S + Hh | −0.083 | −0.136 | 0.018 |
| | S + Dh | −0.083 | −0.168 | 0.027 |

**Table 3.** Correlation analysis between aboveground biomass, abiotic factors, and soil organic carbon in each soil layer.

| Depth of Soil Laye | Aboveground Biomass | Altitude | Slope Degree | Total Nitrogen | Total Phosphorus | Total Potassium | pH |
|---|---|---|---|---|---|---|---|
| 0–10 | 0.596 ** | 0.239 * | 0.190 | 0.936 ** | 0.132 | −0.274 * | −0.080 |
| 10–20 | 0.105 | 0.474 ** | 0.038 | 0.915 ** | 0.179 | −0.185 | −0.122 |
| 20–30 | 0.207 | 0.537 ** | 0.004 | 0.915 ** | 0.131 | −0.178 | −0.277 * |

Notes: * means *p* < 0.05; ** means *p* < 0.01.

The effects of driving factors on soil organic carbon in different soil layers was shown in Table 4. For 0–10 cm soil layer, abiotic factors had both direct (0.985) and indirect effects (−0.021) on soil organic carbon, species diversity and structural diversity also had direct (−0.036, 0.184) and indirect effects (0.002, 0.015) on soil organic carbon, but the direct effects of the three factors on soil organic carbon was more prominent, while the aboveground biomass had only a direct positive effect (0.188) on soil organic carbon. For soil layer 10–20 cm, abiotic factors had both direct (0.98) and indirect effects (−0.015) on soil organic carbon, but the direct was stronger than the indirect effects, and species diversity had both direct (−0.03) and indirect effects (−0.005) on soil organic carbon, while structural diversity

had only direct negative effects (−0.04), but the two effects on soil organic carbon were extremely low.

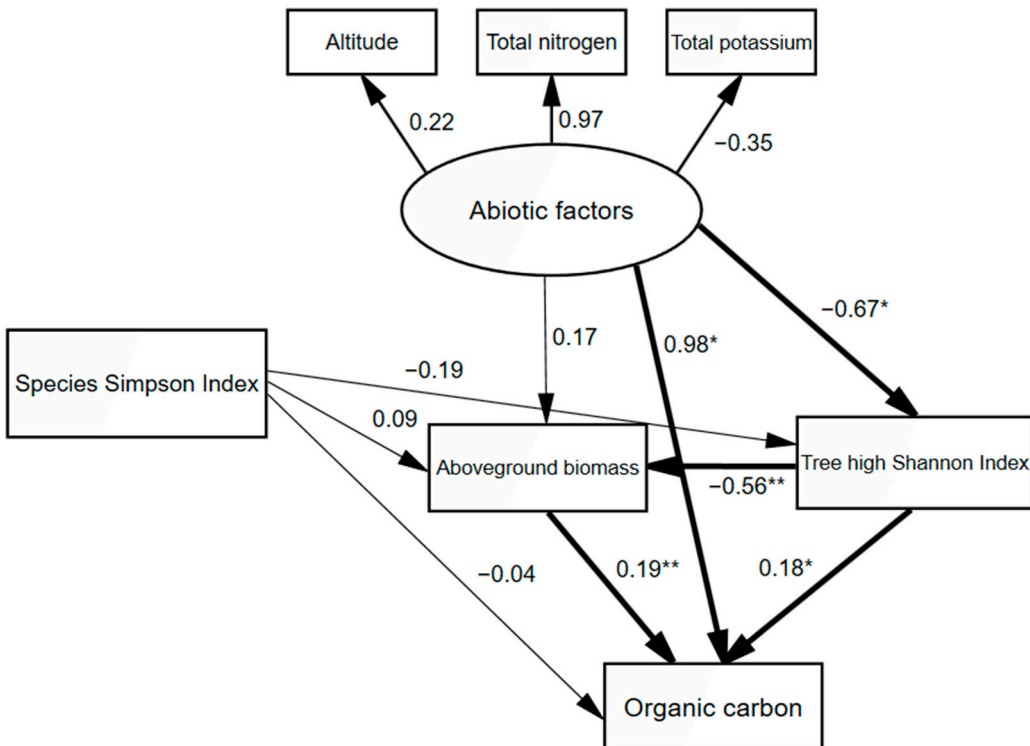

**Figure 2.** Optimal structural equation model of driving factors of soil organic carbon in 0–10 cm. (Notes: Use thick and thin arrows to show the significance of causality among variables. Among them, the bold arrow indicates that the causal relationship between the two variables is significant; the thin line indicates that it is not significant; * $p < 0.05$ and ** $p < 0.01$).

**Table 4.** Standardization effect of driving factors of soil organic carbon in each soil layer.

| Depth of Soil Laye | Predictor | Direct Effect | Indirect Effect | Total Effect |
|---|---|---|---|---|
| | Abiotic factors | 0.985 | −0.021 | 0.964 |
| 0–10 | Species Simpson index | −0.036 | 0.002 | −0.034 |
| | Tree height Shannon index | 0.184 | −0.015 | 0.169 |
| | Aboveground biomass | 0.188 | — | 0.188 |
| | Abiotic factors | 0.98 | −0.015 | 0.965 |
| 10–20 | Species Shannon index | −0.03 | −0.005 | −0.035 |
| | DBH Simpson index | −0.04 | — | −0.04 |

In conclusion, there was no significant correlation between species diversity and soil organic carbon content in typical stands in the western mountainous areas of Beijing ($p > 0.05$). Structure diversity and aboveground biomass had only a significant effect on organic carbon in the surface soil (0–10 cm) and played a dominant role in the driving mechanism of organic carbon in the surface soil. The effects of abiotic factors on organic carbon in all soil layers were significantly correlated, but they played a predominant role in the driving mechanism of organic carbon in the subsurface layer (10–20 cm) and deep layer (20–30 cm), and there were both direct and indirect effects on organic carbon in all soil layers, while the direct effects were stronger than the indirect effects.

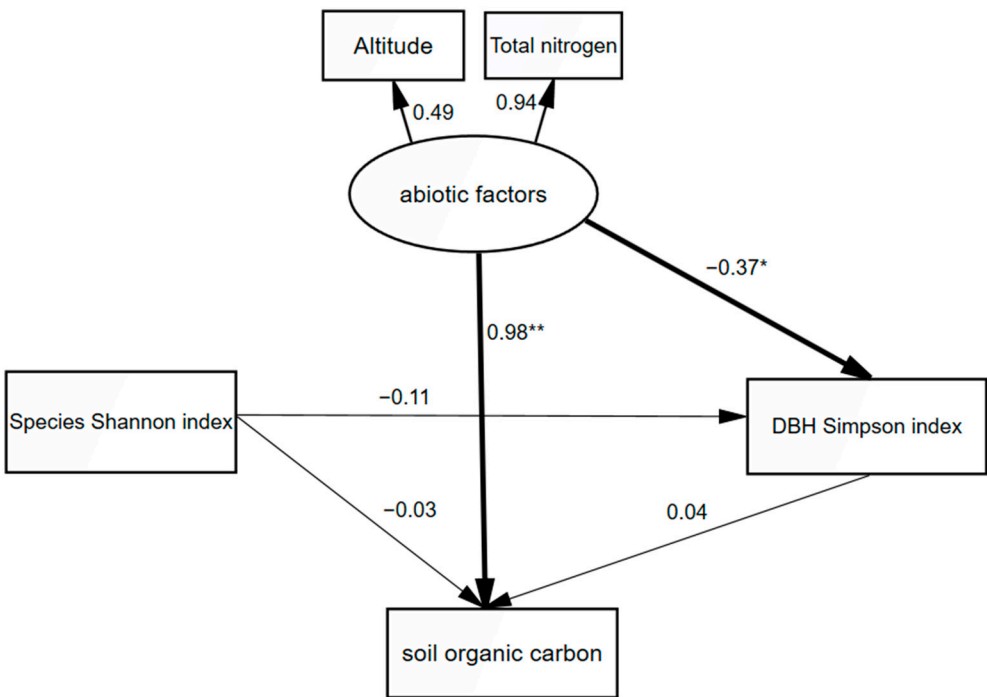

**Figure 3.** Optimal structural equation model of driving factors of soil organic carbon in 10–20 cm. (Notes: Use thick and thin arrows to show the significance of causality among variables. Among them, the bold arrow indicates that the causal relationship between the two variables is significant; the thin line indicates that it is not significant; * $p < 0.05$ and ** $p < 0.01$).

## 4. Discussion

Exploring the driving mechanism of soil organic carbon has always been one of the core and hot issues in ecological research. It was of great significance theoretical to clarify the driving factors of soil organic carbon for sustainable forest management. As a statistical method based on confirmatory factors analysis and path analysis to reveal a certain phenomenon, the structural model has a great advantage in simulating and verifying complex relationships among multiple factors, and has been widely used in forestry and ecology research fields. For example, Wei et al. [27] used the structural equation model to study the influence of regional topography and stand structure on soil characteristics; Hou et al. [28] used the structural equation model to study the optimal vegetation cover and planting slope to control soil erosion during rainfall; which research results fully reflected the scientificity and reliability of the model. In this study, species diversity, structural diversity, and aboveground biomass were used to characterize biotic factors, and abiotic factors were characterized by altitude, soil total nitrogen, and soil total potassium, the direct and indirect effects of biotic and abiotic factors on organic carbon in each soil layer were discussed by structural equation model. The results showed that the interaction between biotic (aboveground biomass and structural diversity) and abiotic factors (altitude and total nitrogen) commonly affected the formation of soil organic carbon. Among them, biotic factors (aboveground biomass and structural diversity) were the main contributors to soil surface organic carbon, and species diversity had no significant effect on soil organic carbon. This is basically consistent with the research conclusion of You et al. [29] thought that the aboveground biomass and litter reserves were the main driving forces affecting the organic carbon of surface soil, it may be that the aboveground biomass directly determines the existing amount of litter in the soil surface layer, and the decomposition of litter accelerates the formation of organic carbon in the soil surface layer [24]. On the other hand, the larger the aboveground biomass, the larger the fine root biomass and carbon inputs of underground vegetation, which accelerates the formation of organic carbon in the soil surface layer [30,31]. However, with the increase in soil depth, the density of plant root

exudates and litter gradually decreases, which makes the inputs of soil organic carbon by aboveground plants decrease; therefore, the biotic factors had no significant impact on organic carbon content in the subsurface and deep soil. At the same time, it was found that the diversity of community structure was also an important biotic factor affecting the formation of soil organic carbon, and it had a negative correlation with soil organic carbon. It may be that the complexity of community structure directly determines the utilization efficiency of soil nutrient resources by plants, and the structural diversity has an inhibitory effect on the aboveground biomass so that the inputs of soil organic carbon and nitrogen into the ground are inhibited, and then the content of soil organic carbon gradually decrease [32,33]. However, Xu et al. [34] thought that abiotic factors played a predominant role in driving the organic carbon in the soil surface layer compared with the structural characteristics of vegetation communities, which is different from the conclusion of this study. This may be the differences in climate types, site conditions, tree species composition, stand structure attributes, and other factors in the study sample plots. Therefore, we could pay more attention to the analysis of the influence of multiple factors on soil organic carbon in future research.

It was found that abiotic factors had relatively little influence on the surface soil organic carbon but played a dominant role in the formation of subsurface and deep soil organic carbon. Among them, total nitrogen and altitude were the main contributors to the formation of soil organic carbon in the subsurface and deep layers and had a positive correlation with soil organic carbon content. This is basically consistent with the conclusion of Zhao et al. [35] thought that altitude was the main driving force for the formation of soil organic carbon and was positively correlated with altitude. It is different from the research conclusions of Sheikh et al. [36] and Liu et al. [37] thought that soil organic carbon increases and then decreases with elevation. This may be the difference caused by factors such as climate and stand structure type of the study sample plot. A large number of studies have shown that the change of altitude gradient includes comprehensive changes in water, heat, light, and other environmental factors, which directly or indirectly change the physical and chemical properties of soil under the forest and the growth of plants [38]. Wei et al. [27] also thought that altitude had a significant impact on stand structure and soil characteristics. We found that the influence of altitude on soil organic carbon not only had a direct effect, but also changed the aboveground biomass and the diversity of community structure by adjusting the temperature, light, and water necessary for the growth and development of aboveground plants [39], and indirectly changes the content of organic carbon inputs into the soil by plants [40]. On the other hand, a series of environmental changes such as water, heat, and light changed the turnover of soil nutrients, and soil microorganisms indirectly affected the formation of soil organic carbon [41]. Drought at low altitudes limited the growth of plants and microbial activities, and moderate- and high-altitude areas had an appropriate combination of temperature and precipitation, which was more conducive to the formation of soil organic carbon [42]. We also found that abiotic factors were related to the formation of organic carbon in surface soil (0–10 cm), but their influence intensity was extremely low. This may be because the existing amount of litter in the surface soil was large, and the microbial activity was strong in Beijing mountainous area; the extremely significant correlation between aboveground biomass and community structure diversity may cover up the influence of abiotic factors on the accumulation of organic carbon in the surface soil. At the same time, it was found that the formation of soil organic carbon was closely related to the total nitrogen content, and with the increase in total nitrogen content, the organic carbon content also increased, which was basically consistent with the conclusion of Liu et al. [43] thought that total nitrogen played a leading role in the formation of soil organic carbon, and the organic carbon content increases with the increase in total nitrogen content; Lu et al. [44] also thought that nitrogen deposition accelerated the carbon fixation efficiency of tropical forest soil, On the one hand, this may be that the increase in nitrogen content greatly reduced the leaching loss of soil organic carbon, which may prevent microbial mineralization of soil organic carbon by changing the nutrient

requirements of microorganisms, and then inhibit the discharge of dissolved organic carbon below the root zone [45]. On the other hand, the increase in nitrogen content significantly changes the physical interaction between the surface of soil fine particles and organic matter decays and then affects the formation of soil organic carbons [41,46]. However, Wen et al. [46] thought that soil pH was also the dominant factor driving the formation of soil organic carbon. This study thought that pH may indirectly affect the excitation effect by changing the community structure of microorganisms, thus alleviating the decomposition of organic carbon; while the study had not reached a similar conclusion, it may be the differences in the pH range of the study plots.

Reasonable stand structure was the key to improving the service function of the forest ecosystem. While the landform conditions in mountainous areas of Beijing were rather bad, how to improve the diversity of forest plants and carbon storage was of great significance to the afforestation project of the Beijing-Tianjin sandstorm source. This study found that the diversity of community structure and aboveground biomass were the dominant biotic factors affecting the organic carbon in surface soil, but the diversity of community structure inhibited the formation of soil organic carbon, while the aboveground biomass promoted the formation of soil organic carbon. Therefore, the aboveground biomass and community structure can be properly adjusted to improve the forest soil organic carbon storage. In this study, when using the structural equation model, we did not fully consider whether climate variables such as air temperature and precipitation would affect the formation of soil organic carbon. Therefore, in future research, we could consider obtaining more data on environmental variables, increasing the number of sample plots, and comprehensively and systematically analyzing the real mechanism that affects the formation of soil organic carbon in woodland.

## 5. Conclusions

In the western mountainous areas of Beijing, the driving factors affecting soil organic carbon vary with the depth of soil layers. Biotic factors (aboveground biomass and structural diversity) were the main contributors to the formation of organic carbon in the soil surface layer (0–10 cm). Among them, there was only a direct positive effect of aboveground biomass on soil organic carbon, while there were both direct and indirect effects of community structure diversity on soil organic carbon, and the direct effect played a major role. Abiotic factors (altitude, total nitrogen, and total potassium) played an important role in the formation of organic carbon in the topsoil (0–10 cm), but they were not the main influencing factors. Biological factors had no significant effect on the formation of soil organic carbon in the subsurface (10–20 cm) and deep (20–30 cm) layers, while abiotic factors played a dominant role in the formation of organic carbon in the subsurface and deep soil, and there were both direct and indirect effects, but the direct effect played a major role. Therefore, the community structure and aboveground biomass can be adjusted appropriately to improve the soil organic carbon content of forest land, taking appropriate measures such as artificial nitrogen deposition to increase the carbon storage of the forest ecosystem.

**Author Contributions:** Conceptualization, methodology, formal analysis, investigation, resources, P.Z. and L.Z.; validation, P.Z., L.Z. and S.Q.; software, data curation, writing—original draft preparation, writing—review and editing, visualization, P.Z.; project administration, supervision, funding acquisition, S.Q. All authors have read and agreed to the published version of the manuscript.

**Funding:** This study was funded by the Beijing-Tianjin Sandstorm Source Control Phase II Project Forestry Monitoring and Evaluation in 2020, Grant No. 2020-SYZ-01-17JC05, and APC was funded by the Beijing-Tianjin Sandstorm Source Control Phase II Project Forestry Monitoring and Evaluation Project in 2020. Evaluation, Grant No. 2020-SYZ-01-17JC05. This project is a cooperation project between Beijing Forestry University and Beijing Municipal Forest and Parks Bureau undertaken by Professor Qi Shi and implemented in Beijing.

**Institutional Review Board Statement:** Not applicable.

**Informed Consent Statement:** Not applicable.

**Data Availability Statement:** The data presented in this study are available on request from the corresponding author.

**Conflicts of Interest:** The authors declare no conflict of interest.

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
