# Peer review of "Plant Diversity and Aboveground Biomass Interact with Abiotic Factors to Drive Soil Organic Carbon in Beijing Mountainous Areas"

_sustainability, doi:10.3390/su141710655_

Round 1

Reviewer 1 Report

Review of Sustainability manuscript 1838509 – “Plant diversity and aboveground biomass interact with abiotic 2 factors to drive soil organic carbon in Beijing mountainous areas.”

Manuscript Recommendation

This manuscript addresses key research questions concerning soil organic carbon (SOC) accumulation and distribution in forested ecosystems of Beijing province China. The field methods and data collection are scientifically valid, and results receive appropriate statistical analysis The research data are adequately presented. The English text, however, is awkward, wordy, and uses redundant phrases. The manuscript needs major revision to address this limitation. As an example, the abstract is rewritten as to improve written clarity. In addition, some key items in the Material and methods need greater explanation and detail and a few format errors exist in tables. Detailed comments follow that outline the needed edits.

Example Abstract

Abstract: We analyzed and compared effects of biotic factors (species diversity, structural diversity, aboveground biomass) and abiotic factors (topography, soil properties) on soil organic carbon in selected mountainous areas of Beijing province China. An overall goal is to provide a preliminary scientific basis for biodiversity protection and coordinated development of forest ecosystems and the subsequent carbon balance in this region. Our study ecosystems were coniferous forests, mixed coniferous and broadleaved forests, and broadleaved forests in the western mountainous area of Beijing. We determined both direct and indirect effects of abiotic and biotic factors on soil organic carbon using multiple linear regression and a structural equation model. Results showed that the biotic factors aboveground biomass and structural diversity were the main driving forces of organic carbon accumulation in the soil surface layer (0−10 cm), but species diversity had no significant effect (P > 0.05). Abiotic factors (altitude, total nitrogen, and total potassium) had some influence on soil surface organic carbon but was subordinate to the biotic factors. The biotic factors had no significant effect on soil organic carbon in the subsurface layers (10−20 cm) and (20−30 cm) (P > 0.05). Whereas the abiotic factors altitude and total nitrogen played a dominant role in subsurface soil organic carbon accumulation of both layers (P < 0.01). The influences were both direct and indirect influences. But the direct influences played a major role. Our results form an initial reference for afforestation management (Beijing-Tianjin Sandstorm-source Control Project) from the perspective of biodiversity protection and carbon storage enhancement.

Detailed review comments.

 Materials and Methods

A figure with a map should be added to show the general location of the study areas within Beijing province and within China. Beijing is both a city and province name thus a map is essential.

 Materials and Methods

Greater explanation is needed on the soil analysis methods and sample preparation. What was the total number of samples collected? How were the soil samples prepared? Air dried? Crushed and sieved at 2 mm?

How was SOC and other analyses determined? Cite the methods used and provide references.

Soil texture has a well-established influence on carbon content. Was soil texture measured?

Table 2

The 10-20 cm depth appears to be on the wrong line. It should be one line up with HS+HD variable.

Table 3

This table has two columns labelled Total Potassium each with different correlation coefficients. Is on of these columns another variable? Explain or correct.

The PH in the last column should be pH.

Citations and References

In the reference section, each reference is numbered (1-45); citations in the text, however, employ author names instead of reference numbers. MDPI publications use a numbered citation style.

Reviewer 2 Report

sustainability-1838509: This manuscript is interesting and suitable for this journal. However, some issues must be improved as the comments and suggestions below:

Lines 44- 46: Factors influencing SOC must be more focused on SEA areas. Please see these papers. [2020. Factors controlling soil organic carbon sequestration of highland agricultural areas in the Mae Chaem Basin, Northern Thailand. Agronomy, 10, 305.] [2008. Variation of soil organic carbon estimates in mountain regions: A case study from Southwest China. Geoderma 146, 449–456.]

Line 99-108: The map of study area is required. Topography and the location of study sites should be presented.

Line 111: “…human disturbance and better growth condition…” This is the important information. Please explain more details. What kind of human disturbance? Why the better growth condition is in June?

Lines 118-121: How many soil samples were taken?

Lines 122-123: What kind of soil chemical and physical properties were measured? By which method?

Line 123: What do you mean “other indicators”?

Table 3: Altitude and slope degree are reported in the results. The method/tool for measurement must be mentioned in Methodology.

Table 3: “Depth of soil laye” must be “Depth of soil layer”

Table 3: “PH” must be “pH”

Table 3: Why you presented two columns of “Total potassium”? It must be some mistake.

All references style in the text were wrong. It must be “[1]” or “[1-4]” .

Round 2

Reviewer 1 Report

Review 2 of Sustainability manuscript 1838509 – “Plant diversity and aboveground biomass interact with abiotic factors to drive soil organic carbon in Beijing mountainous areas.”

Manuscript Recommendation

Authors have added a location figure and address format and addressed labeling errors in tables. In the cover letter response authors also provide information about soil sample number, sample preparation, and lab analyses. The sample and analysis information needs to be incorporated into the Methods and materials and references for analytical methods should be provided. An example paragraph for soil sample and analyses is provided in the detail comments that follow. A few possible references for analytical methods are also included. The English text remains awkward and wordy in sections and needs revision. The first paragraph from the introduction is redone as an example to improve written clarity.

Detailed comments

Introduction

Line 30

The statement -- Soil represents the largest carbon pool of forest ecosystem—is questionable.

Forest ecosystems may contain more C in above ground biomass than is contained is SOC.

For example, See this reference

Tang, X., Zhao, X., Bai, Y., Tang, Z., Wang, W., Zhao, Y., Wan, H., Xie, Z., Shi, X., Wu, B. and Wang, G., 2018. Carbon pools in China’s terrestrial ecosystems: New estimates based on an intensive field survey. Proceedings of the National Academy of Sciences, 115(16), pp.4021-4026.

Lines 30-46

Example rewrite to improve clarity.

Soil is a major carbon pool in forest ecosystems. Soil carbon fluctuates due to abiotic (e.g., climate, soil properties, topography) and biotic (e.g., vegetation density and diversity). Thus, a better understanding of soil carbon sequestration is an important aspect in the global carbon cycle and climate change modification. [1-4]. Key to our understanding of soil organic carbon formation and accumulation is how plant diversity and above ground biomass influence subsurface carbon [5,6]. Soil organic carbon is intricately regulated through dynamics such as plant root exudates [7], root contribution [8], litter decomposition[9] and competition between plants and decomposers[10]. Recent small-scale studies have shown that plant diversity can increase SOC storage by increasing underground carbon inputs,[11] increasing microbial activity, and inhibiting carbon loss during decomposition[12]. Alternatively, several large-scale studies have demonstrated that formation and storage of SOC was regulated by various agents including climate,[13] topography, vegetation types, and other influences[14,15]. Therefore, how biotic factors (plant diversity, above ground biomass) and abiotic factors drive formation of soil organic carbon in forest ecosystems warrants further study.

Methods and materials

The soil sample and analyses information provided in the author response needs to be incorporated into the Methods and Materials and appropriate references cited when possible.

For an example

Each layer of soil samples was repeated for three times producing a total of 432 samples. Sample preparation involved air drying, mixing, crushing, and sieving (2 mm).  Laboratory analysis included organic carbon, total nitrogen, total potassium, total phosphorus, and pH,. Total nitrogen soil was determined by the Kjeldahl analysis via infrared N furnace.  Total phosphorus was determined from a HCIO4-H2SO4 dissolution by spectrophotometer. Total potassium was determined was determined from a HNO3-HClO4 digestion by atomic absorption. Soil pH was determined in 1:1 soil water suspension. Organic carbon was determined by the Walkley Black method.

The following are possible references for soil analyses like those described by the authors.  These may or may not be the correct references in this manuscript depending on the exact procedures and instrumentation used.

Bremner, J.M., 1965. Total nitrogen. Methods of soil analysis: part 2 chemical and microbiological properties, 9, pp.1149-1178.

 Pratt, P.F., 1965. potassium. Methods of Soil Analysis: Part 2 Chemical and Microbiological Properties, 9, pp.1022-1030.

 Sommers, L.E. and Nelson, D.W., 1972. Determination of total phosphorus in soils: a rapid perchloric acid digestion procedure. Soil Science Society of America Journal, 36(6), pp.902-904.

 Walkley, A. and Black, C.A., 1965. Organic carbon. Methods of soil analysis. American Society of Agronomy, Madison, 37, pp.1372-1375.

Author Response

Thank you for your careful review. For detailed interpretation, please refer to the uploaded revision instructions and manuscript

Reviewer 2 Report

Some minor issues need further revision.

1) Lines 108-121: “A total of 432 soil samples” must be reported in the paper.

2) Line 121: “total nitrogen, total potassium, total phosphorus and pH value” must be replaced the “other indicators”.

3) All the number references in the text should not be in the superscript style. Please revise all.

4) Lines 396-399: These references must be presented as below:

Arunrat, N.; Pumijumnong, N.; Sereenonchai, S.; Chareonwong, U. Factors controlling soil organic carbon sequestration of highland agricultural areas in the Mae Chaem Basin, Northern Thailand. Agronomy 2020, 10(2), 305.

Zhang, Y.; Zhao, Y.C.; Shi, X.Z.; Lu, X.X.; Yu, D.S.; Wang, H.J.; Sun, W.X.; Darilek, J.L. Variation of soil organic carbon estimates in mountain regions: A case study from Southwest China. Geoderma 2008, 146, 449–456.

Author Response

(The authors gave the same response as above.)
